# Serum Amino Acid Profiles in Dogs with a Congenital Portosystemic Shunt

**DOI:** 10.3390/metabo15040258

**Published:** 2025-04-09

**Authors:** Robert Kyle Phillips, Amanda B. Blake, Michael S. Tivers, Alex Chan, Patricia E. Ishii, Jan S. Suchodolski, Jörg M. Steiner, Jonathan A. Lidbury

**Affiliations:** 1Gastrointestinal Laboratory, Texas A&M University, 4474 TAMU, College Station, TX 77843-4474, USA; 2Paragon Veterinary Referrals, Paragon Point, Red Hall Crescent, Wakefield WF1 2DF, UK; 3Bristol Vet Specialists, Unit 10, More Plus Central Park, Madison Way, Severn Beach, Bristol BS35 4ER, UK

**Keywords:** amino acid profiling, canine, portosystemic shunt, branched-chain amino acids, aromatic amino acids, Fischer’s ratio, surgical attenuation, shunt morphology, multivariate analysis

## Abstract

**Background/Objectives**: A functional liver is vital for normal protein metabolism. Alterations of circulating amino acid (AA) concentrations have previously been reported in dogs with hepatocellular carcinoma, chronic hepatitis, and hepatocutaneous syndrome. The purpose of this study was to compare serum AA profiles between dogs with a congenital portosystemic shunt (CPSS) and healthy control dogs. **Methods**: Serum samples were collected from 50 dogs with an extrahepatic congenital portosystemic shunt (eCPSS) and 10 dogs with an intrahepatic congenital portosystemic shunt (iCPSS) at time of surgical intervention and from 21 healthy control dogs. Serum AA and other nitrogenous compounds were measured with a dedicated amino acid analyzer. The concentration of each AA was compared between groups using a Kruskal–Wallis test followed by Dunn’s multiple comparisons tests, as appropriate. The Benjamini–Hochberg procedure was used to control for false discovery. Significance was set at q < 0.05. **Results**: Compared to healthy controls, dogs with a CPSS had significantly increased serum concentrations of ammonia, asparagine, glutamic acid, histidine, phenylalanine, serine, and tyrosine and had significantly decreased concentrations of isoleucine, leucine, threonine, urea, and valine. There were no significant differences in serum AA concentrations between dogs with an eCPSS and dogs with an iCPSS. **Conclusions**: Dogs with a CPSS had altered serum AA concentrations compared to healthy control dogs, including decreased branched-chain amino acids (BCAAs) and increased aromatic amino acids (AAAs). In summary, serum AA profiles can differentiate dogs with a CPSS from healthy dogs but not dogs with an eCPSS from dogs with an iCPSS.

## 1. Introduction

The liver plays a central role in the metabolism of many essential macromolecules, including lipids, carbohydrates, and proteins. Diseases that affect liver function by causing alterations to normal anabolic and catabolic pathways may elicit measurable changes in the serum concentrations of various metabolites, such as serum amino acids. In human patients with liver disease (namely, cirrhotic patients with hepatic encephalopathy), evaluation of plasma amino acid concentrations, especially the ratio of the concentrations of branched-chain amino acids (BCAAs) to aromatic amino acids (AAAs), has been studied as a proxy for monitoring hepatic parenchymal damage [1,2,3]. Similar changes in amino acid profiles have been reported in dogs with various liver diseases [4,5,6,7].

Congenital portosystemic shunts (CPSSs) are developmental vascular abnormalities that allow blood from the portal circulation to directly enter the systemic circulation without first being processed by the liver [8]. Depending on their location relative to the liver, congenital shunts may be described as either intrahepatic (iCPSSs) or extrahepatic (eCPSSs). In dogs, iCPSSs are more commonly seen in larger breeds, while eCPSSs are more commonly seen in smaller breeds [9]. Portosystemic shunting may result in hepatic encephalopathy (HE), which is associated with a constellation of neurological signs including lethargy, listlessness, depression, head pressing, ataxia, seizures, and even coma [10]. While ammonia dysmetabolism is central in the pathogenesis of HE, other factors such as systemic inflammation, altered neurotransmitters, or oxidative stress may also play a synergistic role [11].

For both iCPSSs and eCPSSs, the current recommended treatment is attenuation of the aberrant vessel with the goal of restoring portal blood flow through the liver [12,13]. Improvement in clinical signs following successful shunt attenuation is thought to reflect a return to more normal hepatic functionality; however, normalization of all measurable biochemical parameters (including amino acids) may lag behind the resolution of clinical signs [14,15,16,17].

The primary goal of the present study was to compare serum amino acid concentrations between dogs with iCPSSs or eCPSSs and healthy control (HC) dogs. Secondarily, we aimed to compare serum amino acid concentrations of CPSS dogs at the time of surgery and at a later follow-up appointment. We hypothesized that serum AA profiles differ between dogs with CPSS and HC dogs and that those differences in AA concentrations would improve following surgical treatment.

## 2. Materials and Methods

### 2.1. Animals

Dogs presenting to the Langford Vets Small Animal Referral Hospital (University of Bristol, Bristol, UK) between July 2015 and September 2018 for surgical treatment of a congenital portosystemic shunt (CPSS dogs) were prospectively enrolled in this observational study. CPSS dogs were split into two subgroups based on shunt type: extrahepatic (eCPSS) or intrahepatic (iCPSS). The diagnostic criteria for CPSSs were the identification of a portosystemic shunt on intra-operative mesenteric portovenography and visual identification of the shunting vessel during an exploratory celiotomy. Depending on vessel morphology and the patient’s tolerance to occlusion, shunts were treated with a gradual attenuation device (e.g., ameroid constrictor or cellophane band) or with complete or partial suture ligation. Healthy control (HC) dogs owned by faculty and staff at Texas A&M University were enrolled in November 2018. The health of these dogs was determined by the absence of clinical signs based on an owner questionnaire; physical examination; and the lack of clinically relevant abnormalities on a complete blood count, gastrointestinal panel (i.e., serum quantification of pancreatic lipase immunoreactivity [cPLI, as measured by Spec cPL], trypsin-like immunoreactivity [cTLI], cobalamin, and folate), serum biochemistry profile, and fecal microbiota dysbiosis index [18]. Basic demographic data including breed, sex, and age were collected from both groups of dogs. Samples were collected from dogs enrolled after obtaining their owners’ informed consent following institutionally approved animal use protocols (AWERB—University of Bristol, VIN/15/037B; IACUC—Texas A&M University, 2017-0190 CA).

### 2.2. Medical Management of CPSSs Prior to Surgery

Prior to undergoing surgical attenuation, most CPSS dogs received various medical interventions (for differing durations). This management almost always included some combination of change in diet with the addition of lactulose, generally some sort of antibiotic therapy, and rarely the administration of steroids. Response to medical treatment was categorized as follows: “resolved” if the dog’s clinical signs fully resolved; “improved” if the dog showed some improvement but not complete resolution of clinical signs; and “unchanged” if the dog had only limited or no improvement at all in clinical signs following medical management. Details of any medical management continued following shunt correction were not available.

### 2.3. Evaluation of Hepatic Encephalopathy (HE)

The severity of HE in CPSS dogs at the time of surgery was graded by one of the current study’s authors (M.S.T.) following a 4-point grading scheme previously described [13,19]. Briefly, dogs receiving a score of “1” were considered normal, showing no abnormal clinical signs. Dogs that showed mild symptoms, such as being lethargic, apathetic, minimally disorientated, or having subtle changes in personality or exhibiting inappropriate behavior, were scored “2”. Dogs that showed moderate signs including hypersalivation, loss of muscle control, drowsiness while still being responsive to verbal commands, head pressing, and circling received a score of “3”. Finally, dogs received a score of “4” when presenting with severe signs, namely, repeated seizures, near-unconsciousness, or coma.

### 2.4. Sample Collection and Processing

Fasted blood samples were collected from CPSS dogs before surgical attenuation for routine diagnostic purposes. An additional follow-up blood sample was available for a small number of dogs who later underwent repeat surgery or returned for post-operative examination. Residual serum not needed for clinical purposes was used for the current study. Serum from all CPSS dogs was aliquoted following routine centrifugation of clotted blood and was immediately transferred to a −80 °C freezer before being shipped on dry ice to the Gastrointestinal Laboratory at Texas A&M University (College Station, TX, USA) in September 2018. These samples and fasted serum from HC dogs remained at −80 °C until AA analysis in December 2020 [20,21].

### 2.5. Amino Acid Analysis

Amino acids and other nitrogenous compounds were identified and measured using a Biochrom 30+ lithium high performance amino acid analyzer (Biochrom Ltd., Cambridge, UK), a commercially available instrument based on ion exchange chromatography with post-column derivatization of samples using ninhydrin. Sample preparation followed a previously analytically validated protocol [21]. Briefly, serum samples were deproteinized in microcentrifuge tubes by a 1:1 (*v*/*v*) addition of 5% sulfosalicylic acid and 500 µM L-norleucine (serving as an internal standard). After vigorous vortexing, tubes were centrifuged at 10,000 *rcf* for 5 min at 4 °C. Up to 500 µL of supernatant was transferred to a 0.2 µm PVDF centrifugal filter tube and again centrifuged at 10,000 *rcf* for 5 min at 4 °C. The resulting filtrate was stored at −80 °C until analysis.

Deproteinized CPSS and HC samples were randomized and placed on the instrument’s chilled (4 °C) autosampler in batches of 20 or fewer in order to prevent samples from remaining on the autosampler for longer than 48 h prior to being injected onto the column. Injection volume was set to 30 µL. Quantification of individual analytes was accomplished by integrating the area-under-the-curve (AUC) of individual peaks (normalized against AUC of the norleucine peak) compared against a known set of standards.

Concentrations of the following amino acids and nitrogenous compounds were determined: alanine (Ala), ammonia (Amm), arginine (Arg), asparagine (Asn), citrulline (Citr), glutamine (Gln), glutamic acid (Glu), glycine (Gly), histidine (His), isoleucine (Ile), leucine (Leu), lysine (Lys), methionine (Met), phenylalanine (Phe), proline (Pro), serine (Ser), taurine (Taur), threonine (Thr), tryptophan (Trp), tyrosine (Tyr), urea, and valine (Val). Additionally, total branched-chain amino acids (BCAAs; Val + Ile + Leu), total aromatic amino acids (AAAs; Phe + Tyr), Fischer’s ratio [22] (BCAAs/AAAs), and BCAAs/Tyr ratio (BTR) were calculated for each dog.

### 2.6. Statistical Analyses

Shapiro–Wilk tests and visual inspection of q-q plots characterized the data as generally having a non-normal distribution. Individual amino acid concentrations were compared among groups using a Kruskal–Wallis test followed by Dunn’s multiple comparisons tests, as needed. Wilcoxon matched-pairs signed-rank tests were used to compare AA concentrations at the time of surgery with concentrations at follow-up appointments. Where significantly different, AA concentrations at follow-up were compared with those of HC dogs using Mann–Whitney tests. The Benjamini–Hochberg procedure was used to control the false discovery rate due to multiple comparisons, and significance was set at q < 0.05. Univariate analyses were performed using Prism v.8.3.0 for Windows (GraphPad Software, San Diego, CA, USA).

Following log transformation and mean-centered scaling of the data, multidimensional analyses comparing all amino acid concentrations across groups were performed using a publicly accessible web platform (MetaboAnalyst v.5.0, available at https://metaboanalyst.ca, as accessed on 7 November 2023) [23]. Output from principal component analysis (PCA) [24], partial least squares discriminant analysis (PLS-DA) [25], and hierarchical cluster heatmaps [26] was used to visualize the clustering of samples and to determine which amino acids made the greatest contribution towards group differentiation. Permutational multivariate analysis of variance (PERMANOVA) was used to test the hypothesis that the centroids and dispersion of groups were equivalent [27].

## 3. Results

### 3.1. Study Population

A total of 60 dogs with a congenital portosystemic shunt (CPSS) were enrolled, including 50 eCPSS dogs and 10 iCPSS dogs. Additionally, 21 HC dogs were enrolled. Basic descriptive characteristics of healthy control dogs versus CPSS dogs are summarized in Table 1.

The population of CPSS dogs included 14 mixed-breed dogs. The most commonly represented purebred dog breeds (count) included pug (10), Yorkshire terrier (8), shih tzu (4), Jack Russell terrier (3), West Highland white terrier (3), Labrador retriever (2), and miniature schnauzer (2). The remaining purebred dogs included a single dog of each of the following breeds: Bernese mountain dog, bichon frisé, border collie, border terrier, cavalier King Charles spaniel, dachshund, golden retriever, Irish setter, Maltese, Norfolk terrier, Nova Scotia duck tolling retriever, Pomeranian, Staffordshire bull terrier, and vizsla. The population of HC dogs included seven mixed-breed dogs. The most commonly represented purebred dog breeds (count) included Labrador retriever (4) and miniature dachshund (2), followed by a single dog of each of the following breeds: American cocker spaniel, Catahoula leopard dog, German shepherd, golden retriever, Great Dane, greyhound, Jack Russel terrier, and Staffordshire bull terrier.

HC dogs were significantly older than dogs in either the eCPSS group (*p* = 0.0298) or the iCPSS group (*p* = 0.0038). Fewer CPSS dogs tended to be spayed/neutered compared with HC dogs (37% and 81%, respectively). Overall, the sex of the dogs was fairly evenly split between male (47%) and female dogs (53%). All other comparisons between groups were not significantly different (*p* > 0.05).

Biochemical panels were performed on CPSS dogs prior to surgical intervention. As summarized in Appendix A, there were no statistically significant differences between dogs with an eCPSS and dogs with an iCPSS for the five parameters investigated: ALT, albumin, BUN, cholesterol, and glucose. Correlations between serum biochemistry analytes and AA concentrations are presented in Appendix A. Very few significant correlations were present, which suggests that changes in the concentrations of a single AA do not recapitulate current liver functional assessments. Serum chemistries were also run on blood samples from HC dogs at the time of study enrollment, but the results were not directly comparable to the results from CPSS dogs due to differences in the chemistry analyzer platforms that were utilized.

### 3.2. Efficacy of Medical Management Protocols

Most CPSS dogs were transitioned to a prescription hydrolyzed diet (63%) or liver diet (27%) for a median duration of 31 days (range, 5 to 1095 days) prior to surgery. Antibiotics were administered orally in 54 out of 60 dogs for a median duration of 33 days (range, 5 to 1209 days) before surgery with amoxicillin–clavulanate (58%) and amoxicillin (25%) being most common. All but one CPSS dog received oral lactulose. Response to medical management was generally good with most dogs showing either an improvement (52%) or a full resolution (32%) of clinical signs. Moreover, at the time of surgery, most dogs presented with no evidence of HE (77%) or only mild signs (15%). These data are summarized in Appendix A.

### 3.3. Amino Acid Concentrations

Twenty AAs and two nitrogenous compounds were measurable in serum samples. Additionally, total AAAs, total BCAAs, Fischer’s ratio, and BTR were calculated using the assayed concentrations. The median and range data for each analyte by group are summarized in Appendix A. The concentrations of 6 AAs (i.e., Asn, Glu, His, Phe, Ser, and Tyr) and Amm were significantly increased in eCPSS and iCPSS dogs compared with HC dogs. The concentrations of 4 AAs (i.e., Ile, Leu, Thr, and Val) and urea were significantly decreased in eCPSS and iCPSS dogs compared with HC dogs. Ten AAs (i.e., Ala, Arg, Citr, Gln, Gly, Lys, Met, Pro, Taur, and Trp) were not significantly different in eCPSS and iCPSS dogs compared to HC dogs.

When comparing AAs in aggregate, total AAAs were significantly increased in eCPSS and iCPSS dogs and total BCAAs were significantly decreased in eCPSS and iCPSS dogs compared with HC dogs. Fischer’s ratio and BTR were also significantly decreased in eCPSS and iCPSS dogs compared with HC dogs.

For all comparisons between eCPSS and iCPSS dogs only, there were no significant differences in serum AA concentrations, as shown in Figure 1.

### 3.4. Effect of Shunt Attenuation on AA Concentrations

An additional serum sample was available for 15 dogs who were seen for follow-up examination after surgery. The median elapsed time since surgery was 97 days, ranging from 78 days to 251 days. Of these follow-up visits, 13 were for eCPSS dogs and 2 were for iCPSS dogs. Six dogs (40% of follow-ups) would go on to require repeat surgical intervention. Seven AAs (i.e., Ile, Leu, Lys, Phe, Thr, Tyr, and Val), two nitrogenous compounds (i.e., Amm and urea), and all four calculated metrics (i.e., total BCAAs, total AAAs, Fischer’s ratio, and BTR) were significantly changed from the baseline (at the time of surgery) following surgical attenuation. When the median follow-up concentrations were compared to those of HC dogs, only Amm remained significantly different. Comparisons of all analyte concentrations between baseline (i.e., at time of first surgery) and follow-up appointment are listed in Appendix A and presented graphically in Figure 2. When assessing the effect of time since surgery (Appendix A) or the effect of need for repeat surgery (Appendix A), no clear inferences could be drawn, and no formal statistical tests were performed.

### 3.5. Multivariate Analyses

The PCA plot demonstrated a fairly tightly packed clustering of HC dogs that was separate from the broader overlapping regions of eCPSS and iCPSS dogs (Figure 3A). The total variance of the first two principal components contributed 56.0% in the PCA model for all three groups (PC 1 = 36.8% and PC 2 = 19.2%). PERMANOVA testing revealed a significant difference among groups (*p* = 0.001). To optimize the separation between groups, this dataset was subjected to PLS-DA (Figure 3B). HC dogs showed a distinct separation from iCPSS dogs and a minimal overlap with eCPSS dogs. Most iCPSS dogs fell within the shaded area of the eCPSS dogs. In the PLS-DA model, the first two components together accounted for 47.0% of total variance. The variables that provided for the best separation in the PLS-DA were ranked by their variable importance in projection (VIP) scores (Figure 3C). The top 10 variables that provided the greatest discriminatory power among the three groups were BTR, Fischer’s ratio, Amm, Tyr, Thr, Glu, total AAAs, Phe, Ile, and Leu. Finally, to better visualize if there was any pattern to the clustering of these variables across groups, a hierarchical cluster heatmap was generated (Figure 3D). When analyzing trends in AA concentrations across groups, notable bands of relative extreme increases or decreases (visualized as darker reds or blues, respectively) were generally more appreciable in comparisons of HC dogs with iCPSS dogs than with eCPSS dogs.

## 4. Discussion

The results of this prospective study demonstrated that some, but not all, serum amino acid concentrations were significantly different when dogs with a CPSS were compared with HC dogs. We found that total aromatic amino acids (AAAs), as well as Phe and Tyr individually, were significantly increased in CPSS dogs compared with HC dogs. Conversely, total branched-chain amino acids (BCAAs), as well as Ile, Leu, and Val individually, were significantly decreased. Fischer’s ratio and BCAAs-to-Tyr ratio (BTR) were also significantly decreased in CPSS dogs compared with HC dogs. This recapitulates findings from early studies where experimentally created portacaval shunts in mongrel dogs resulted in elevated AAAs and decreased BCAAs concentrations [4,5]. Additionally, we observed significant increases in concentrations of Asn, Glu, His, and Ser, and a significant decrease in concentrations of Thr in CPSS dogs when compared with HC dogs. In an untargeted analysis of serum metabolites, CPSS dogs showed a similar increase in Ser and decrease in Thr when compared with HC dogs and dogs with chronic hepatitis [6].

For the present study, it is important to note that blood was collected from CPSS dogs at the time of surgery after approximately 1 month of generally successful medical management. As such, the changes in AAs reported here were those that persisted despite clinical improvements. This point reflects the limited nature of medical management of CPSS dogs, where overt symptoms of HE may lessen, but the underlying liver dysfunction is not fully addressed.

Multivariate analysis techniques have increasingly been used to examine group changes in large datasets. Recent examples include studies that aimed to explore the utility of AA profiles to evaluate the effects of glucose supplementation on the plasma metabolome of dogs and cats [28], to associate plasma AA concentrations with the severity of hepatocutaneous syndrome in dogs [29,30], and to compare the serum metabolome of dogs with CPSS to dogs with markedly elevated serum ALT concentrations [7]. In the present study, we used multivariate analysis as an aid for visualizing and understanding the significance of changes (or lack thereof) in 26 parameters (22 measured analytes and 4 calculated metrics). The PCA plot, which evaluated all serum analyte concentrations concurrently, showed that there was considerable overlap between eCPSS and iCPSS dogs. The PLS-DA plot, which attempts to optimize our ability to discriminate between groups, showed that this overlap between eCPSS and iCPSS dogs persisted. Based on the associated VIP scores, we found that BCAAs (Ile, Leu, and Val), AAAs (Tyr and Phe), and especially the calculated ratios incorporating concentrations from both (Fischer’s and BTR) were among the most important factors for differentiating between groups. Amm and Thr were also identified as among the top five discriminating metrics. The heatmap of AA concentrations relative to each group highlights how the concentrations of BCAAs (individually and in aggregate) are consistently the lowest in iCPSS dogs. However, these trends were not significantly different following univariate analyses.

The liver plays a central role in the catabolism of most amino acids with the notable exception of BCAAs, which are primarily catabolized in skeletal muscle. In patients with severe liver disease, serum concentrations of amino acids, including Phe and Tyr, may increase due to diminished hepatic function [31]. Conversely, serum concentrations of BCAAs (i.e., Val, Leu, and Ile) may be normal or even significantly decreased due to the increased utilization of BCAAs for ammonia detoxification [32]. In an effort to sequester excess circulating ammonia arising from reduced urea production within the liver, enzymes in skeletal muscles transfer the amino group from BCAAs to α-ketoglutarate, forming Glu. This newly created Glu can then act as an ammonia sink releasing less toxic Gln back into the bloodstream [33].

Unsurprisingly, two urea cycle analytes differed between HC dogs and dogs with a CPSS, as reflective of their impaired liver function. Dogs with a CPSS had significantly increased serum Amm concentrations and significantly decreased serum urea concentrations when compared with HC dogs. Given the liver’s central role in the conversion of circulating ammonia into urea via the urea cycle, these changes should be expected for animals with a CPSS due to portal blood flow bypassing the liver and entering directly into systemic circulation [10]. Increased serum ammonia concentrations are detrimental to the central nervous system [34] and play a central role in the pathogenesis of hepatic encephalopathy, a common sequela for dogs with a CPSS [35,36]. Two other AAs from the urea cycle that we measured in this study (Arg and Citr) were not statistically different between groups.

### 4.1. Changes in AA Concentrations After Shunt Attenuation

Of the 60 CPSS dogs that initially underwent surgery for shunt attenuation, only 15 dogs returned for a post-operative follow-up appointment, at which time an additional serum sample was drawn and available for analysis. The elapsed time between surgery and follow-up was not standardized, ranging from 78 days to 251 days. In a small study of dogs with experimentally created portocaval shunts, AA profiles were tracked over the course of 28 weeks and showed fairly immediate and lasting changes in BCAA and AAA concentrations following surgery [37]. Our hypothesis was that dogs receiving surgical correction of their CPSS would have AA profiles that would normalize to levels comparable to those in HC dogs. In this study, of the follow-up analyte concentrations that were significantly different from concentrations measured at the time of surgery, only Amm concentrations remained significantly different (increased) compared with HC dogs. All other analytes did not differ significantly from HC concentrations.

These findings augment those published by Devriendt et al. who concluded that most AA concentrations were only partially improved at 3 months post-surgery [17]. Similar to their results, we also observed a decrease in Phe and Tyr concentrations and no change in Trp concentrations following surgery. Where they only saw Val concentrations increase post-surgery, we demonstrated significant increases in concentrations of Val as well as in concentrations of Ile and Leu. Both studies saw no significant changes in post-surgery concentrations of Met or Ser. In contrast to their study, a period greater than 3 months elapsed for some of our dogs prior to follow-up serum being collected. We also did not observe an increase in concentrations of Arg or Citr, nor did we see any significant decrease in concentrations of Gln, Pro, or His following surgery. The median Fischer’s ratio of their 10 CPSS dogs was 0.5 at time of surgery and only improved to 1.5 after three months. In comparison, our 15 CPSS dogs had median Fischer’s ratios of 0.9 at time of surgery and improved to 3.5 after a median of 97 days following surgery. As such, we consider the possibility that the dogs in our study were less severely affected at the time of surgery.

We found 17 AAs in our study that were not significantly different following surgery. For the 13 AAs that did change significantly between surgery and follow-up, the general trend observed was for AA concentrations to improve back to levels approaching those of HC dogs. One outlier visually (but not statistically) was Lys, the concentration of which appeared to rebound following surgery and achieve concentrations greater than those seen in HC dogs. Lysine is an essential amino acid in dogs and is sourced strictly from their diet, which may be fortified in some hepatic care formulations of dog food [17]. Many dogs in our study had been transitioned to a prescription diet (predominantly either a hydrolyzed protein diet or a diet formulated for dogs with liver disease) for a median time of 31 days prior to surgery as part of their medical management. Unfortunately, we do not have complete dietary histories for these dogs post-surgery, so the effect of a possible diet change (if any) on serum amino acid concentrations between the time of surgery and follow-up appointment is unknown.

Some dogs were re-evaluated as early as 78 days or as late as 251 days following surgery. We would have expected that those dogs seen later would have had more time for their AA concentrations to normalize (i.e., for concentrations to become more like the concentrations observed in HC dogs), but this was not the case. As shown in Appendix A, of the five dogs with follow-up appointments greater than 3 months after surgery, four dogs had Fischer’s ratios below the median value for all follow-up dogs. Because it is possible that the reason these dogs re-presented to the hospital was, in fact, due to them not improving at a similar pace to other dogs, as reflected in their AA concentrations, no meaningful conclusions can be easily drawn. Similarly, we considered the possibility that the median AA concentrations of our follow-up results may have been skewed toward the levels seen at surgery due to the inclusion of dogs with partial surgical attenuations. However, as demonstrated in Appendix A, of the six dogs that ultimately required repeat surgery to fully correct their CPSS, dogs were split between those with Fischer’s ratios below (n = 4) and above (n = 2) the median value for all follow-up dogs. Accordingly, the extent to which the need for repeat surgery affected AA concentrations at follow-up remains unclear. Future studies with rechecks of all dogs (both those doing well and those not doing well) at set timepoints that evaluate the resolution of post-operative clinical signs and that systematically record AA concentrations monthly for a course longer than 3 months (e.g., up to 1 year after shunt attenuation) should be undertaken. This could provide a more complete picture of liver functional restoration over time.

In human medicine, studies of cirrhotic patients that tracked amino acids normalization following liver transplantation generally have shown promising results. One study investigating predictors of post-operative sepsis found that Fischer’s ratios and concentrations of BCAAs and Gln were already significantly improved at day 7 after liver transplantation [38]. An earlier study that looked at amino acids in both human patients and dogs who had undergone liver transplantation found that AAAs returned to within the normal range from their preoperative concentrations within the first day after transplantation and that Fischer’s ratios had largely stabilized in or near the range of normal for most of the duration of the 30-day study [39]. In a longer-term (median 35 months) follow-up of human patients receiving liver transplantation due to end-stage chronic liver failure, most AA concentrations had normalized by this time with the exception of BCAAs, the concentrations of which were improved but remained significantly lower than those in control subjects [40]. Just as the analysis of serum AAs following liver transplantation has served as a useful marker of liver function in human patients, so too might serum AA profiling play a future role in assessing the restoration of liver function in dogs with a CPSS.

### 4.2. Study Limitations

The present study was limited by the number of iCPSS dogs included. Compared with the number of dogs with an eCPSS (n = 50) and the number of HC dogs (n = 21), the number of dogs with an iCPSS (n = 10) was considerably smaller. That said, other recent studies have been published which included similar numbers of dogs [7,17,30]. Had we been able to enroll more iCPSS dogs, some of the trends in the differences in AA concentrations observed between eCPSS and iCPSS dogs that were ultimately non-significant (e.g., increased Asn and Phe in iCPSS and decreased Ile, Leu, and Val in iCPSS) might have become significantly different given the additional statistical power. While we did not necessarily expect there to be changes in AA profiles based on shunt morphology, other associated variables such as the size of dogs due to breed may be important.

In a study reviewing the medical records of 53 dogs with an eCPSS, shunt morphology (e.g., splenocaval vs. splenophrenic) had a significant influence on certain blood parameters (e.g., serum albumin concentrations, total serum protein concentrations, and hematocrit), as well as on the presence of clinical, neurologic, and urinary tract signs [41]. The nature of an individual dog’s shunt (as it relates to the amount and location of liver parenchyma receiving adequate perfusion) may similarly affect serum AA concentrations. For the present study, we only compared dogs with intrahepatic versus extrahepatic shunts, and we did not evaluate the effects of more specific shunt morphology classifications due to limited numbers of each subtype.

Most dogs undergoing surgical attenuation first underwent medical therapy for at least one month. These treatments included changes in the diet (to a prescription diet designed for dogs with liver disease or to a prescription hydrolyzed protein diet), antibiotic therapy, and/or lactulose administration to mixed degrees of success. These efforts, while often varied in implementation and duration, are largely designed to reduce blood ammonia concentrations [42], which may ameliorate some clinical signs but do not appear to significantly affect serum AA concentrations in the near term [17]. This variability in presurgical treatment history (especially diet) of the dogs at the time of sample collection is another limitation of our study. We also do not have records of what medical management regimens the dogs followed after shunt attenuation. Any changes in AA concentrations attributable to the corrective surgery may be confounded by the effects of continued administration of lactulose and/or antibiotic treatments during this time. Although the modulation of serum AA concentrations by lactulose or antibiotics has not been described in dogs, it stands to reason that compounds altering the gut microbiome may elicit changes in host metabolism besides the intended reduction in ammonia [43].

While it is true that most CPSS dogs were receiving a prescription diet as part of their medical management, the diet used was not standardized to a single brand or formulation. As an attempt to mitigate the effect of diet on AA concentrations, serum was collected from dogs after withholding food for at least 12 h in hopes that AA concentrations reached steady state in the absence of active digestion and absorption. However, despite heterogenous diets, robust changes in AA profiles have been observed in human patients with liver disease and in models in other species. Future studies could attempt to limit variability by feeding a single diet to all CPSS dogs.

We acknowledge the limitation of using HC dogs in this study that were neither breed- nor age-matched. As CPSSs are more common in certain breeds, our study design could have been improved had we exclusively made comparisons between like animals (e.g., including only Yorkshire terriers in both our healthy and diseased groups). We do, however, feel that our control group was reasonably heterogeneous with respect to its breed inclusion and age representation, which should help mitigate any breed-related bias. Like many other recent studies investigating amino acid profiles in dogs across disease groups [44,45,46,47], this study chose to compare concentrations in dogs with a contemporaneously recruited healthy control group because, as Tamura et al. states, “There are no accurate species-specific reference ranges for plasma amino acid concentrations in dogs” [48]. In the future, it would be good to establish such common reference intervals for serum amino acid concentrations in dogs. These would ideally adequately capture the differences appreciated across different breeds, sexes, and life stages.

In the disease state, the liver’s ability to adequately metabolize amino acids may become impaired. While diseases affecting the liver are varied (e.g., mediated by inflammatory, neoplastic, metabolic, pathogenic, or genetic processes), very little is known regarding how AA profiles are altered for a specific hepatopathy or stage of disease. By noting the patterns of AA concentrations that seem most affected versus those which seem to be less affected, a better understanding of the underlying pathophysiology of liver diseases might be gleaned. Additional research is needed to better connect the mechanisms of AA metabolism with different hepatic dysfunctions.

## 5. Conclusions

When compared to healthy control dogs, dogs with congenital portosystemic shunts had altered serum AA concentrations, including increased Asn, Glu, His, Phe, Ser, Tyr, and AAA concentrations; decreased Ile, Leu, Thr, Val, and BCAA concentrations; and decreased BCAA-to-AAA ratio and BTR, important metrics for evaluating hepatic encephalopathy in cirrhotic patients. Perhaps more importantly, following surgical attenuation, many of these same amino acids could be seen trending towards concentrations observed in HC dogs, a tendency that should be explored over longer standardized follow-up durations. In this study, we demonstrated that serum AA profiling may serve to highlight the hepatic functional deficits present in dogs with a CPSS. We were unable, however, to distinguish dogs with an eCPSS from dogs with an iCPSS by AA profiling alone. The use of AA profiles as a tool for ongoing prognostic assessment of liver health following surgical treatment of a CPSS deserves further investigation.

## Figures and Tables

**Figure 1 metabolites-15-00258-f001:**
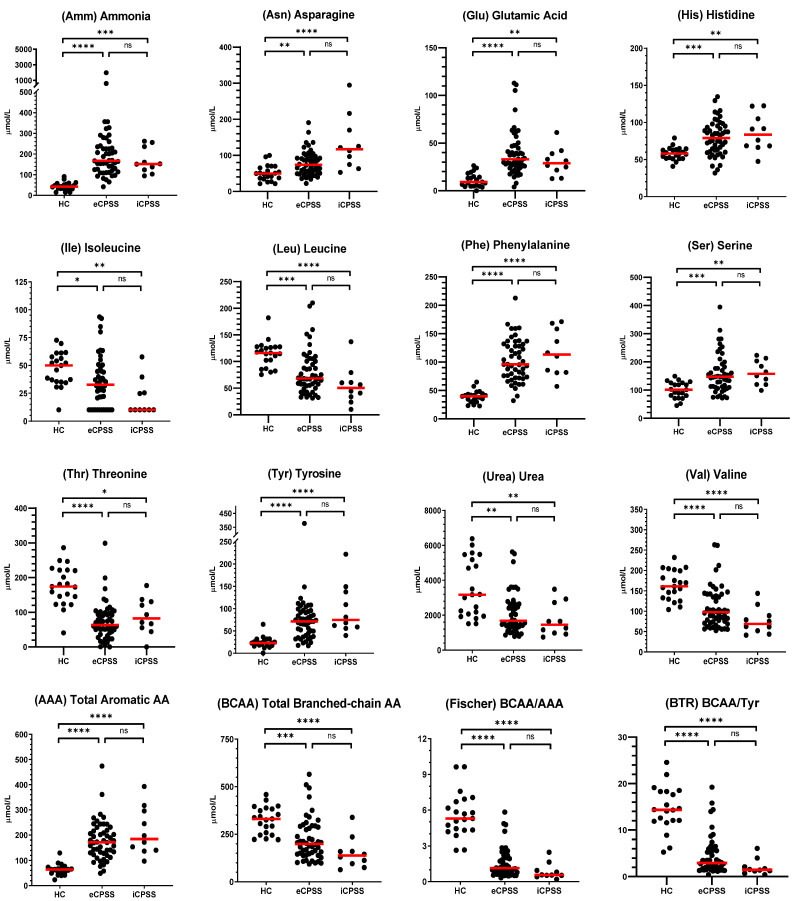
Serum amino acids (AA), nitrogenous compounds, and calculated amino acids metrics (i.e., total AAAs, total BCAAs, Fischer’s ratio, and BTR) for which concentrations were significantly different (at q < 0.05) in dogs with an extrahepatic congenital portosystemic shunt (eCPSS) or an intrahepatic congenital portosystemic shunt (iCPSS) compared with healthy control (HC) dogs. The red lines indicate the median for each group. The asterisks (*, **, ***, or ****) represent a significant difference (at *p* < 0.05, *p* < 0.01, *p* < 0.001, or *p* < 0.0001, respectively) and “ns” represents no significant difference following Dunn’s multiple comparisons test.

**Figure 2 metabolites-15-00258-f002:**
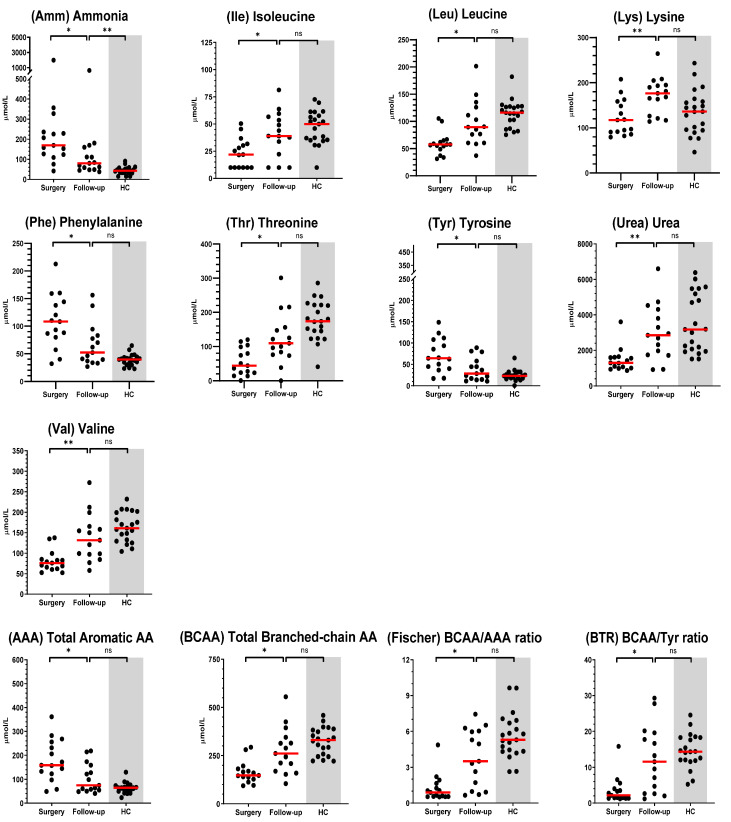
Serum amino acids (AA), nitrogenous compounds, and calculated amino acids metrics (i.e., total AAAs, total BCAAs, Fischer’s ratio, and BTR) for which concentrations were significantly different (at q < 0.05) in dogs between timepoints: at surgery and at follow-up appointment. Concentrations of healthy control dogs (HC, shaded in grey) were subsequently compared to follow-up concentrations. The median for each group is indicated by the red line. The asterisks (*, **, ***, or ****) represent a significant difference (at *p* < 0.05, *p* < 0.01, *p* < 0.001, or *p* < 0.0001, respectively), and “ns” represents no significant difference.

**Figure 3 metabolites-15-00258-f003:**
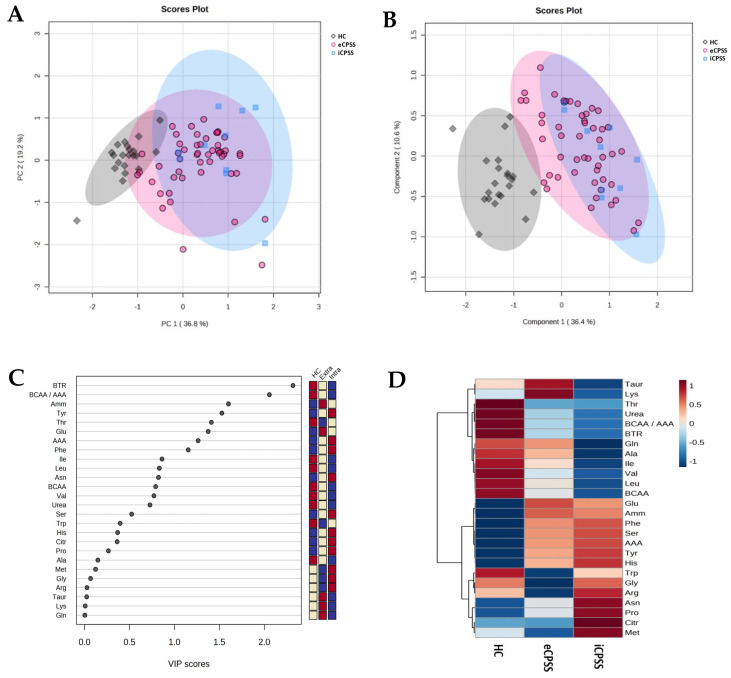
Multivariate analyses of measured analyte concentrations in healthy control dogs (HC; black diamonds, n = 21), dogs with an extrahepatic portosystemic shunt (eCPSS; red circles, n = 50), and dogs with an intrahepatic portosystemic shunt (iCPSS; blue squares, n = 10): (**A**) scores plot of principal component analysis (PCA) and (**B**) scores plot of partial least squares discriminant analysis (PLS-DA). The shaded area represents the 95% confidence interval. The percentages of variance of the data explained by the first and second principal components (PC 1 and PC 2, respectively) are provided in parentheses along each axis. (**C**) Variable importance in projection (VIP) scores of component 1 of the PLS-DA identifying the top 20 discriminating parameters in descending order of importance. The colored cells on the right indicate the relative abundance of each variable across groups: higher values are shown in red, and lower values are shown in blue. (**D**) Hierarchical cluster heatmap of variables. The columns represent the average for each group (HC, eCPSS, and iCPSS). The colored cells indicate the relative abundance of each variable across the groups: higher values are shown in red, and lower values are shown in blue. The horizontal and vertical black lines depict the clustering of parameters.

**Table 1 metabolites-15-00258-t001:** Signalment of healthy control dogs and dogs with an eCPSS or iCPSS. Age is reported as median [and range].

Variable			HC		eCPSS		iCPSS	
Number (n =)			21		50		10	
Age (years)			3.0 [0.6–10.0]		1.1 [0.3–7.3]		0.7 [0.4–3.0]	
Sex	Total Male		10 (48%)		22 (44%)		5 (50%)	
	MI	MN	2	8	16	6	5	0
	Total Female		11 (52%)		28 (56%)		5 (50%)	
	FI	FS	2	9	16	12	1	4
Breed	Mixed		7/21 (33%)		11/50 (22%)		3/10 (30%)	
	Purebred		14/21 (67%)		39/50 (78%)		7/10 (70%)	

Abbreviations: HC, healthy control; eCPSS, extrahepatic congenital portosystemic shunt; iCPSS, intrahepatic congenital portosystemic shunt; MI, male intact; MN, male neutered; FI, female intact; FS, female spayed.

## Data Availability

The data presented in this study are available either within this article or as Appendix A.

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
