# Peer review of "Serum Amino Acid Profiles in Dogs with a Congenital Portosystemic Shunt"

_metabolites, 2025, doi:10.3390/metabo15040258_

Round 1

Reviewer 1 Report

Comments and Suggestions for Authors

The manuscript was examined in detail and the following results were obtained.
The aim of this study was to compare the serum AA profiles between dogs with congenital portosystemic shunt (CPSS) and healthy control dogs.
The abstract was written in a simple and understandable manner and no deficiencies were identified.
Authors can add to line 29. To conclude, It can be said that serum AA profiles can………..
The introduction section was written in a simple way that the reader can easily understand and supported by literature.
In line 100. Add after 2.3. Evaluation of Hepatic Encephalopathy (HE)
In line 287. Earease dogs. Should be compared with.
In line 346. Devriendt et al. [17] should be.
The subject of the study is a current and future topic that needs to be investigated. The number of N was found to be low in the study. In such clinical studies, the high number of N can better determine the changes in dogs with and without congenital portosystemic shunt (CPSS) in which breed, age, gender or body size. However, in the literature searches, it was determined that there were studies with similar numbers of N to this study. Therefore, it was determined that the number of N in the current study was usable.

Author Response

The manuscript was examined in detail and the following results were obtained. The aim of this study was to compare the serum AA profiles between dogs with congenital portosystemic shunt (CPSS) and healthy control dogs. The abstract was written in a simple and understandable manner and no deficiencies were identified.

  • Authors can add to line 29. To conclude, It can be said that serum AA profiles can……….. The introduction section was written in a simple way that the reader can easily understand and supported by literature.
    • Added the words “In summary,” to the beginning of the last sentence of the abstract.
  • In line 100. Add after 2.3. Evaluation of Hepatic Encephalopathy (HE)
    • Added “(HE)” to the end of the subheading, although the abbreviation was previously introduced in lines 50-51.

  • In line 287. Earease dogs. Should be compared with.
    • Re-wrote lines 291–293 to read: “The results of this prospective study demonstrated that some, but not all, serum amino acid concentrations were significantly different when dogs with CPSS were compared with HC dogs.”

  • In line 346. Devriendt et al. [17] should be. The subject of the study is a current and future topic that needs to be investigated. The number of N was found to be low in the study. In such clinical studies, the high number of N can better determine the changes in dogs with and without congenital portosystemic shunt (CPSS) in which breed, age, gender or body size. However, in the literature searches, it was determined that there were studies with similar numbers of N to this study. Therefore, it was determined that the number of N in the current study was usable.
    • Discussion of the limited number of enrolled dogs is provided in the first paragraph of section 4.2. To clarify that our results are relevant despite a small sample number we have added the following new sentence at line 428:  “That said, other recent studies have been published with similar numbers of dogs {Devriendt 2021}{Imbery 2022}{Leela-arporn 2025}.”

Reviewer 2 Report

Comments and Suggestions for Authors

Dear Authors,

This manuscript examines the serum amino acid profiles in dogs with congenital portosystemic shunt (CPSS). The aim is to determine the amino acid differences between healthy dogs and dogs with CPSS and to observe the changes after surgical intervention.
Methodologically, 50 extrahepatic (eCPSS) and 10 intrahepatic (iCPSS) CPSS dogs and 21 healthy control dogs were included in the study. Serum samples were taken and examined with a high-performance amino acid analyzer. Kruskal-Wallis test, Dunn’s multiple comparison test and Benjamini-Hochberg correction were used for statistical analyses.
The manuscript, Although metabolic changes have been investigated in some portosystemic shunt studies before, this study fills an important gap by providing a comprehensive analysis of the amino acid profile. It confirms that aromatic amino acids (AAA) are increased and branched chain amino acids (BCAA) are decreased in dogs with CPSS, thus clarifying the metabolic effects of this disease. It shows that parameters such as Fischer ratio (BCAA/AAA) and BCAA/Tyrosine ratio (BTR) can be used in the evaluation of dogs with CPSS. It emphasizes the necessity of biochemical follow-up after treatment by showing that some amino acids normalize after surgery in dogs with CPSS, but critical metabolites such as ammonia do not completely recover. This manuscript fills an important gap in the field of veterinary metabolomics and hepatology. The new biomarkers it presents, the necessity of metabolic follow-up after surgery, and multidisciplinary analysis techniques may shed light on future research in veterinary medicine and even human medicine.

However, as a major revision, reviewing the following points would add value to the study.
- The average follow-up period after surgery is 97 days (ranging from 78-251 days). Longer follow-up (e.g. 6 months - 1 year) should have been performed to see when hepatic function completely normalized.
- Breed factor could have been taken into account. CPSS is more common in some breeds (e.g. Yorkshire Terrier, Maltese). This may affect the results as the AA profile can naturally vary between breeds.

The manuscript identifies AA changes but does not explain the underlying mechanisms.
Why are the decreases in BCAAs and increases in AAAs significant? More could have been explained regarding possible metabolic pathways (e.g. urea cycle, neurotransmitter synthesis). Serum AA analysis could have been compared with other biomarkers (e.g. ammonia, ALP, ALT).
The paper reports decreased branched-chain amino acids (BCAA: valine, isoleucine, leucine) and increased aromatic amino acids (AAA: tyrosine, phenylalanine) in dogs with CPSS. However, it does not explain the molecular causes of these changes.

Author Response

This manuscript examines the serum amino acid profiles in dogs with congenital portosystemic shunt (CPSS). The aim is to determine the amino acid differences between healthy dogs and dogs with CPSS and to observe the changes after surgical intervention. Methodologically, 50 extrahepatic (eCPSS) and 10 intrahepatic (iCPSS) CPSS dogs and 21 healthy control dogs were included in the study. Serum samples were taken and examined with a high-performance amino acid analyzer. Kruskal-Wallis test, Dunn’s multiple comparison test and Benjamini-Hochberg correction were used for statistical analyses.

The manuscript, Although metabolic changes have been investigated in some portosystemic shunt studies before, this study fills an important gap by providing a comprehensive analysis of the amino acid profile. It confirms that aromatic amino acids (AAA) are increased and branched chain amino acids (BCAA) are decreased in dogs with CPSS, thus clarifying the metabolic effects of this disease. It shows that parameters such as Fischer ratio (BCAA/AAA) and BCAA/Tyrosine ratio (BTR) can be used in the evaluation of dogs with CPSS. It emphasizes the necessity of biochemical follow-up after treatment by showing that some amino acids normalize after surgery in dogs with CPSS, but critical metabolites such as ammonia do not completely recover. This manuscript fills an important gap in the field of veterinary metabolomics and hepatology. The new biomarkers it presents, the necessity of metabolic follow-up after surgery, and multidisciplinary analysis techniques may shed light on future research in veterinary medicine and even human medicine.

However, as a major revision, reviewing the following points would add value to the study.

  • The average follow-up period after surgery is 97 days (ranging from 78-251 days). Longer follow-up (e.g. 6 months - 1 year) should have been performed to see when hepatic function completely normalized.
    • Discussion of this limitation is provided in the paragraph beginning at line 390. We have added verbiage better framing the extent of the optimal long-term follow-up period (lines 405–408):  “Future studies with rechecks of all dogs (both those doing well and those not doing well) at set timepoints that evaluate resolution of post-operative clinical signs and that systematically record AA concentrations monthly for a longer course than 3 months (e.g., up to 1 year after shunt attenuation) should be undertaken.”

  • Breed factor could have been taken into account. CPSS is more common in some breeds (e.g. Yorkshire Terrier, Maltese). This may affect the results as the AA profile can naturally vary between breeds.
    • We concur that more studies are needed to investigate how demographic/genetic factors influence baseline serum amino acid concentrations. Additional commentary was added to the next-to-last paragraph of the discussion section (lines 469–474) addressing these concerns.

  • The manuscript identifies AA changes but does not explain the underlying mechanisms. Why are the decreases in BCAAs and increases in AAAs significant? More could have been explained regarding possible metabolic pathways (e.g. urea cycle, neurotransmitter synthesis).
    • Added paragraph to discussion (line 329–338) discussing BCAA vs. AAA metabolism.
    • Added sentence (line 347–349) to the existing paragraph discussing urea cycle analytes highlighting that two AAs from the urea cycle, Arg and Citr, were not significantly different between groups.

  • Serum AA analysis could have been compared with other biomarkers (e.g. ammonia, ALP, ALT).
    • Table S5 was added to the supplementary file, providing correlations between amino acid concentrations and the serum biochemistry analytes for which we had available data (ALT, albumin, ammonia, cholesterol, and glucose).
    • Added mention of table S5 to line 197–200 of the main text.

  • The paper reports decreased branched-chain amino acids (BCAA: valine, isoleucine, leucine) and increased aromatic amino acids (AAA: tyrosine, phenylalanine) in dogs with CPSS. However, it does not explain the molecular causes of these changes.
    • New paragraph (line 329–338) added with discussion of AAA vs BCAA metabolism.

Reviewer 3 Report

Comments and Suggestions for Authors

Dear authors,

it is positive that this manuscript “Serum Amino Acid Profiles in Dogs with a Congenital Portosystemic Shunt” (metabolites-3551755) by coauthors (Robert Kyle Phillips, Amanda B. Blake , Michael S. Tivers , Alex Chan , Patricia E. Ishii , Jan S. Suchodolski , Jörg M. Steiner , Jonathan A. Lidbury) devoted to the evaluation of the amino acid (AA) profiles in blood serum for various dogs “with a congenital portosystemic shunt (CPSS) and healthy control dogs”. It is positive that the following samples and methods were used: serum samples were collected from 50 dogs with an extrahepatic congenital portosystemic shunt (eCPSS) and 10 dogs with an intrahepatic congenital portosystemic shunt (iCPSS) at time of surgical intervention and from 21 healthy control dogs (serum AA and other nitrogenous compounds were measured with a dedicated amino acid analyzer). The following important results were obtained: Compared to healthy controls, dogs with CPSS had significantly increased serum concentrations of ammonia, asparagine, glutamic acid, histidine, phenylalanine, serine, and tyrosine, and had significantly decreased concentrations of isoleucine, leucine, threonine, urea, and valine. There were no significant differences in serum AA concentrations between dogs with eCPSS and dogs with iCPSS. It is especially interesting that the dogs with CPSS had altered serum AA concentrations compared to healthy control dogs, including decreased branched-chain amino acids (BCAA) and increased aromatic amino acids (AAA), as well as the serum AA profiles can differentiate dogs with CPSS from healthy dogs (but not dogs with eCPSS from dogs with iCPSS). The manuscript analyze the literature works in detail and at high level of discussion. I do not doubt the technical quality of the work and feel that there is a sufficient impact on a broader readership to justify publication in the "Metabolites". This topic is in frame of the journal scope, the subject matter is treated in depth. Thus, the present manuscript is actual and important, especially in the field of the development of the evaluation of the AA-profiles and other metabolites in blood serum for various dogs. It is very positive that the authors attached two supplementary files including the raw data (AAinCPSSdogs_FileS1_RawData).

There are no negative comments on this manuscript. The presence of the “Abbreviation” part somewhere in the text can be valuable.

Author Response

It is positive that this manuscript “Serum Amino Acid Profiles in Dogs with a Congenital Portosystemic Shunt” (metabolites-3551755) by coauthors (Robert Kyle Phillips, Amanda B. Blake , Michael S. Tivers , Alex Chan , Patricia E. Ishii , Jan S. Suchodolski , Jörg M. Steiner , Jonathan A. Lidbury) devoted to the evaluation of the amino acid (AA) profiles in blood serum for various dogs “with a congenital portosystemic shunt (CPSS) and healthy control dogs”. It is positive that the following samples and methods were used: serum samples were collected from 50 dogs with an extrahepatic congenital portosystemic shunt (eCPSS) and 10 dogs with an intrahepatic congenital portosystemic shunt (iCPSS) at time of surgical intervention and from 21 healthy control dogs (serum AA and other nitrogenous compounds were measured with a dedicated amino acid analyzer). The following important results were obtained: Compared to healthy controls, dogs with CPSS had significantly increased serum concentrations of ammonia, asparagine, glutamic acid, histidine, phenylalanine, serine, and tyrosine, and had significantly decreased concentrations of isoleucine, leucine, threonine, urea, and valine. There were no significant differences in serum AA concentrations between dogs with eCPSS and dogs with iCPSS. It is especially interesting that the dogs with CPSS had altered serum AA concentrations compared to healthy control dogs, including decreased branched-chain amino acids (BCAA) and increased aromatic amino acids (AAA), as well as the serum AA profiles can differentiate dogs with CPSS from healthy dogs (but not dogs with eCPSS from dogs with iCPSS). The manuscript analyze the literature works in detail and at high level of discussion. I do not doubt the technical quality of the work and feel that there is a sufficient impact on a broader readership to justify publication in the "Metabolites". This topic is in frame of the journal scope, the subject matter is treated in depth. Thus, the present manuscript is actual and important, especially in the field of the development of the evaluation of the AA-profiles and other metabolites in blood serum for various dogs. It is very positive that the authors attached two supplementary files including the raw data (AAinCPSSdogs_FileS1_RawData).

  • There are no negative comments on this manuscript. The presence of the “Abbreviation” part somewhere in the text can be valuable.
    • Added Abbreviations section before References.

Round 2

Reviewer 2 Report

Comments and Suggestions for Authors

Thank you for making the relevant revisions.